# Interleaved Reasoning for Large Language Models via Reinforcement Learning

## Abstract

Long chain-of-thought (CoT) significantly enhances the reasoning capabilities of large language models (LLMs). However, extensive reasoning traces lead to inefficiencies and increased time-to-first-token (TTFT). We propose a novel training paradigm that uses reinforcement learning (RL) to guide reasoning LLMs to *interleave thinking and answering* for multi-hop questions. We observe that models inherently possess the ability to perform interleaved reasoning, which can be further enhanced through RL. We introduce a simple yet effective reward scheme to incentivize correct intermediate steps, guiding the policy model toward correct reasoning paths by leveraging intermediate signals generated during interleaved reasoning. Extensive experiments across five diverse datasets and three RL algorithms (PPO, GRPO, and REINFORCE++) demonstrate consistent improvements over traditional think-answer reasoning, without requiring external tools. Our method improves final task accuracy and overall efficiency by enabling more effective credit assignment during RL. Specifically, our approach reduces TTFT by over 80% on average, reduces overall reasoning length by 37%, and achieves an average 12.5% improvement in final Pass@1 accuracy. Furthermore, our method, trained solely on question answering and logical reasoning datasets, exhibits strong generalization to complex reasoning datasets such as MATH, GPQA, and MMLU. Additionally, we conduct in-depth analysis to reveal several valuable insights into conditional reward modeling.

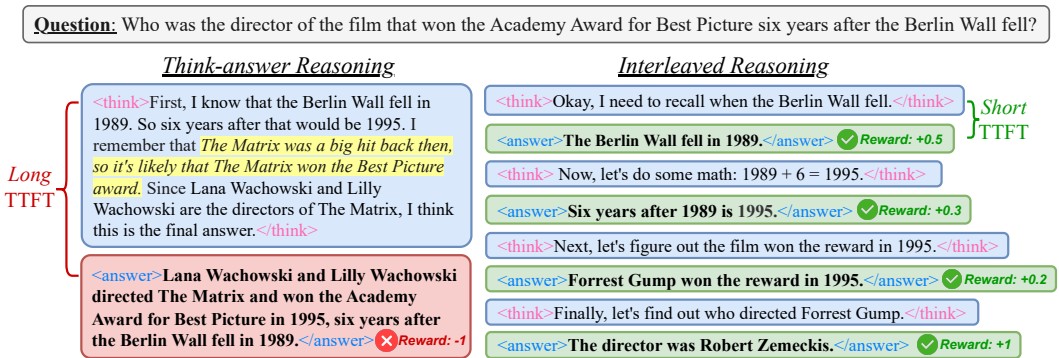

Figure 1: Standard think-answer reasoning (left) completes the full chain-of-thought before generating an answer, resulting in high TTFT and making credit assignment difficult during training when intermediate steps contain errors (highlighted in yellow). Interleaved reasoning (right) alternates between thinking and answering, enabling structured, easy-to-verify reward signals for better credit assignment and significantly reducing TTFT.

## 1 Introduction

Reasoning large language models (LLMs) (Jaech et al., 2024; Guo et al., 2025) have demonstrated advanced capabilities in complex multi-hop tasks through long chain-of-thought (CoT) (Wei et al., 2022). However, the standard *"think-answer"* paradigm, where models must complete the full reasoning trace before generating answers, introduces two critical limitations. *First*, it significantly increases time-to-first-token (TTFT), taking seconds or minutes for answer generation. This breaks the interaction flow in real-time AI applications such as conversational assistants, resulting in poor

user experience. *Second*, by delaying answer generation until the reasoning concludes, models may follow incorrect intermediate steps, propagate errors, and lead to inaccurate final answers and reasoning inefficiencies such as overthinking (Chen et al., 2024; Sui et al., 2025).

Humans naturally provide incremental feedback during conversations, signaling understanding even as they formulate complete responses. Decomposing a complex problem into smaller steps is also the de-facto approach for many reasoning tasks in LLMs (Wei et al., 2022; Khot et al., 2022; Zhou et al., 2022; Besta et al., 2023). However, current reasoning LLMs treat thinking and answering as strictly sequential processes – answers are available only after reasoning concludes.

Current reasoning reinforcement learning (RL) paradigms often treat intermediate reasoning traces as byproducts or unstructured chatter (Kumar et al., 2024; Hou et al., 2025; Guo et al., 2025). However, we argue that for multi-hop reasoning tasks, structured intermediate answers are valuable on several fronts. *First*, unstructured reasoning streams often contain exploratory and potentially contradictory thoughts, and users rarely have the bandwidth to examine such lengthy reasoning traces (Treude and Kula, 2025). Yet these traces may already include partial conclusions that can be useful; clearly presenting such conclusions early can enhance the interaction experience (Liu et al., 2025). *Second*, most production reasoning LLMs do not stream their reasoning content in real time (Comanici et al., 2025; OpenAI, 2025). Making the problem-solving process visible provides transparency and helps users verify the model's final output. *Third*, these partial conclusions can also be utilized as dense supervision signals to further improve model's reasoning during training (Lightman et al., 2023; Cui et al., 2025). Ideally, models should iteratively switch between "think" and "answer" modes based on their understanding of the problem and its complexity. However, effectively applying RL to induce such behavior remains challenging. It is unclear whether models can learn and generalize across various complex tasks. Moreover, effectively leveraging simple, rule-based rewards to detect sufficient intermediate signals during training is largely under-explored.

To address these challenges, we introduce *interleaved reasoning*, a novel RL training method that enables LLMs to interleave thinking and answering. As shown in Figure 1, an interleaved reasoning model generates concrete and informative intermediate answers during reasoning, while providing reward signals for training. We conduct comprehensive experiments on three popular RL algorithms (PPO (Schulman et al., 2017), GRPO (Shao et al., 2024), and REINFORCE++ (Hu, 2025)) and five diverse datasets (K&K (Xie et al., 2024), Musique (Trivedi et al., 2022), MATH (Hendrycks et al., 2021), GPQA (Rein et al., 2023), MMLU (Hendrycks et al., 2020)) and found that LLMs are inherently capable of answering questions in an interleaved manner (§3.2.1). We introduce a simple yet effective reward scheme that provides consistent feedback for intermediate steps during training (§3.2.2), resulting in an average 12.5% Pass@1 improvement in final task accuracy and significantly reducing TTFT by over 80% on average (§5.1). By guiding the model to stay on the correct reasoning path, our method results in up to 37% shorter reasoning length compared to traditional think-answer reasoning and generalizes strongly to unseen and challenging tasks (§5.2). Finally, our comprehensive analysis reveals several valuable and practical insights into reward modeling, stable RL training, and the dynamics of model reasoning.

## 2 RELATED WORK

**LLM Reasoning and Efficiency.** Research on enhancing LLMs' reasoning capabilities has followed several key directions. Early approaches focused on improving base or instruction-tuned LLMs through techniques like chain-of-thought prompting (Wei et al., 2022), self-consistency (Liu et al., 2021), and few-shot learning (Brown et al., 2020), while others explored structured reasoning through graph-based methods (Besta et al., 2023). Another line of work leverages external tools and APIs (Lewis et al., 2020; Gao et al., 2022; Chen et al., 2022) to augment model capabilities. Recent developments in RL enable models like OpenAI-o1 (Jaech et al., 2024) and DeepSeek-R1 (Guo et al., 2025) to generate long CoT to improve their reasoning abilities. This shift towards longer reasoning also results in inefficiencies and significantly increased latency and Time-to-First-Token (TTFT). Recent studies address this issue by proposing more concise reasoning through techniques such as inference-time adjustments (Xu et al., 2025b;a; Kimi, 2025), length control RL (Aggarwal and Welleck, 2025; Fatemi et al., 2025; Yuan et al., 2025), or additional finetuning (Luo et al., 2025). Interleaving reasoning with action using RL is also a newly emerging research area. Concurrent work mainly focuses on leveraging *external* tools such as search engines (Jin et al., 2025; Chen et al.,

2025; Song et al., 2025; Li et al., 2025b) during the reasoning process. In contrast, we focus on the model's *internal* ability to generate concrete intermediate answers, which can later be used as additional reward signals for training.

**Reinforcement Learning for LLM Reasoning.** Currently, reinforcement learning (RL) (Kaelbling et al., 1996) is the dominant approach to convert a base LLM into a reasoning LLM (Kumar et al., 2024; Hou et al., 2025; Guo et al., 2025; Xie et al., 2025), where the model is rewarded based on the correctness of the final answer and adherence to the reasoning format. Reward modeling is a strong means of guiding a model to learn new skills during RL (Silver et al., 2021). There are primarily two types of rewards used during RL: Outcome Reward Models (ORMs) and Process Reward Models (PRMs). DeepSeek R1 (Guo et al., 2025) demonstrates that simple rule-based ORMs can significantly improve performance on challenging reasoning tasks. PRMs are often used to provide denser feedback on intermediate steps (Lightman et al., 2024; Uesato et al., 2022; Wang et al., 2024). However, they face significant practical challenges - they often require human annotation for generated output (Lightman et al., 2024; Uesato et al., 2022), which inevitably introduces risks of reward hacking (Rafailov et al., 2024), requiring training a separate reward model (Wang et al., 2024) and adding complexity to the training pipeline (Guo et al., 2025). In this work, we leverage the concept of PRM, but instead of relying on a separate learned model, we only use a simple rule-based reward to capture intermediate signals. Unlike PRMs that generate feedback at each step during rollout, our method operates more like an ORM while granting partial credit to the intermediate answers. Discussions on the distinction between PRM and our method can be found in Appendix A. We leverage a conditional reward scheme similar to Yuan et al. (2025). However, instead of focusing on reducing response length, our work focuses on improving the quality of intermediate reasoning.

## 3 TRAINING LLMs FOR INTERLEAVED REASONING

In this section, we present our approach for training LLMs to interleave thinking and answering. We first formalize the interleaving process and then describe our reinforcement learning formulation.

### 3.1 PRELIMINARY

We conceptualize answering a multi-hop question as a sequence of resolved intermediate sub-problems. A sub-answer is a user-facing piece of information or partial conclusion that the model confidently derives at a given stage. The model *should* output a sub-answer when it determines that a self-contained part of the problem has been solved or a meaningful milestone in reasoning has been reached. For example, a sub-answer might resolve the first sub-problem and guide the next - such as an intermediate calculation in a multi-hop problem. In this way, the overall response is constructed incrementally through clear and conclusive sub-answers.

**Thinking vs. Answering.** The distinction between thinking and answering requires careful consideration. From a philosophical perspective, thinking constitutes an integral component of answer formulation. However, from a user experience standpoint, a model's answer *begins* when the first valid answer token is generated. Based on their utility to the user, we define *thinking* as a private internal reasoning process that is not accessible or useful to the user. In contrast, *answering* is the generation of public, finalized conclusions that constitute a meaningful response to the user's question. These conclusions may represent partial solutions to the overall problem, but they are presented as complete intermediate steps that advance the user's understanding or problem-solving process.

Formally, given user input $x$ requiring $N$ reasoning steps, the policy model $\pi_\theta$ produces a sequence $y$ that alternates between thinking and answering segments. Let $k \in \{1, \ldots, N\}$ index the steps. We denote the thinking segment by $y_{\text{think}}^{(k)}$ and the corresponding answer segment by $y_{\text{answer}}^{(k)}$. The interleaved generation thus is

$$y = y_{\text{think}}^{(1)} \circ y_{\text{answer}}^{(1)} \circ y_{\text{think}}^{(2)} \circ y_{\text{answer}}^{(2)} \circ \cdots \circ y_{\text{answer}}^{(N)}, \tag{1}$$

where $\circ$ denotes concatenation. The final answer to the original question is $y_{\text{answer}}^{(N)}$, whereas the preceding answer segments $\{y_{\text{answer}}^{(k)}\}_{k=1}^{N-1}$ are intermediate answers. The thinking segments $y_{\text{think}}^{(k)}$ guide the reasoning process but are not part of the user-visible answer for the TTFT calculation until the subsequent answer segment $y_{\text{answer}}^{(k)}$ is produced.

## 3.2 REINFORCEMENT LEARNING FOR INTERLEAVED REASONING

We formulate the task of learning interleaved reasoning as a reinforcement learning problem. During RL, the policy model $\pi_\theta$ generates sequences that maximize an expected reward while maintaining generation quality. The objective function is:

$$\max_{\pi_\theta} \mathbb{E}_{x\sim\mathcal{D}, y\sim\pi_\theta(\cdot|x)} \left[ r(x,y) \right] - \beta D_{\mathrm{KL}} \left[ \pi_\theta(y \mid x) \,||\, \pi_{\mathrm{ref}}(y \mid x) \right], \qquad (2)$$

where $\mathcal{D}$ is the training dataset, $\pi_{\mathrm{ref}}(y \mid x)$ is the reference policy model, $\beta$ is the KL divergence coefficient, and $r(x,y)$ is the reward function. Detailed hyperparameter choices are discussed in Appendix B. We discuss the policy optimization in §4 and compare the performance of different RL algorithms in §5.2. After training, the model should have learned how to dynamically switch between them based on the given task at each step.

**Interleaved Reasoning Template.** To guide the model in adopting the interleaved reasoning process, we use a specific instruction template during training and inference. Following the recent success of DeepSeek-R1 (Guo et al., 2025), we use two special tags (<think></think> and <answer></answer>) to instruct the model to perform reasoning and provide answers within each tag, respectively. We also use the original template proposed in DeepSeek-R1 for think-answer reasoning (Appendix C). The complete interleaved template is shown in Table 1.

Table 1: Template for interleaving thinking and answering. prompt will be replaced with the specific reasoning question during training.

---

You are a helpful assistant. You reason through problems step by step before providing an answer. You conduct your reasoning within <think></think> and share partial answers within <answer></answer> as soon as you become confident about the intermediate results. You continue this pattern of <think></think><answer></answer><think></think><answer></answer> until you reach the final answer. User: prompt. Assistant:

---

### 3.2.1 REWARD DESIGN

To effectively train the model to reason within the interleaved format, we utilize three rewards: the **format** reward assesses whether the interleaved format is correctly followed and properly completed; the **final accuracy** reward evaluates the correctness of the final answer; and the **conditional intermediate accuracy** reward (or intermediate reward) provides additional rewards for correct intermediate answers, applied conditionally based on training progress. Following previous work (Jin et al., 2025), our reward design avoids complex neural reward models, instead focusing on simple rule-based rewards that provide clear and consistent feedback without requiring separate reward model training. We discuss the conditions to apply the intermediate reward in §3.2.2. More details about the rewards can be found in Appendix D.1.

**Models Are Quick Format Learner.** Our initial experiments revealed that models inherently possess the ability to interleave thinking and answering. Base models (without RL training) can generate intermediate answers by directly applying the interleaved template, with some reduced accuracy. Additionally, models rapidly learn the structural format. As illustrated in Figure 2, the format reward for both reasoning methods quickly plateaus, whereas the accuracy reward continues to improve. We also observe that both reasoning methods achieve similar final accuracy reward during training. The finding suggests the main challenge is not stylistic adherence but rather enhancing the quality of their thought processes for different reasoning tasks.

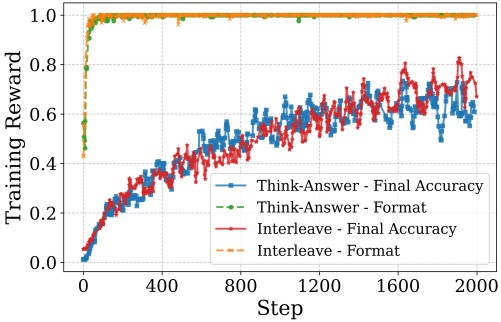

Figure 2: The format reward rapidly reaches a plateau during training, significantly faster than the accuracy reward, suggesting that LLMs naturally adopt structural patterns.

This motivates our focus on the reasoning itself: not for its structure per se, but for its potential to improve the model's reasoning by leveraging its explicit intermediate outputs as learning signals.

### 3.2.2 CONDITIONAL REWARDS

Our finding shows that directly applying intermediate reward during training often leads to suboptimal results, as the model may prioritize local correctness at the expense of final solution correctness (§5.2). To effectively leverage the benefit of intermediate answers beyond shorter TTFT, we design a conditional reward strategy that incentivizes the model to generate correct intermediate answers early, in order to guide the reasoning toward the correct final answer. We apply intermediate rewards only when the model demonstrates foundational competence and shows meaningful learning progress during training. Specifically, the rewards are applied when three conditions are met: (1) the final answer is correct, (2) the output format is valid, and (3) the model shows improvement in the current training batch compared to previous one. The core idea is to ensure that the model first masters the primary objective before optimizing for the sub-tasks of generating correct intermediate steps. Formally, the conditional intermediate reward is defined as:

$$r_{\text{intermediate}}(x, y) \; = \; \mathbb{1}(C) \cdot \sum_{k=1}^{N-1} f\big(y_{\text{answer}}^{(k)}\big), \tag{3}$$

$$\textit{where } C = \text{Format}(y) \; \wedge \; \text{Correct}\big(y_{\text{answer}}^{(N)}\big) \; \wedge \; \big(\text{Acc}(B) > \text{Acc}(B-1) - \epsilon\big), \tag{4}$$

where $\text{Acc}(B)$ denotes the accuracy for the current training batch $B$, $\mathbb{1}(\cdot)$ is the indicator function, $f(y_{\text{answer}}^{(k)})$ evaluates the answer correctness at step $k$, and $f$ is the reward calculation function. Following Yuan et al. (2025), we include $\epsilon$ as threshold for training stability. The batch accuracy criterion serves as a curriculum indicator, gradually introducing intermediate rewards as training progresses. The overall reward function is:

$$r(x, y) = r_{\text{format}}(y) + r_{\text{final}}(x, y) + r_{\text{intermediate}}(x, y), \tag{5}$$

where $r_{\text{intermediate}}(x, y)$ is invoked only if all the aforementioned conditions are met. The full reward definitions can be found in Appendix D.

**Intermediate Reward Calculation.** We explore different approaches to calculate intermediate reward under a conditional nature. While all approaches use the conditional scheme described above, they differ in how they calculate the actual reward value. We explore three approaches: **All-or-None**, which requires all intermediate steps to be correct in sequence; **Partial Credit**, which gives partial credit for individual correct intermediate steps; and **Time-Discounted**, which assigns higher rewards to earlier correct intermediate steps while assigning extra rewards to the all correct intermediate steps. For simplicity, intermediate ground truth are used for the intermediate rewards calculation. However, intermediate answers are *not* a hard requirement for our method (Appendix E). Additionally, despite training *only* on tasks with intermediate ground truths, our method generalizes to other unseen tasks without such annotation and remains effective even *without* intermediate rewards (§5.1). We compare three calculation approaches in §5.2, provide additional details in Appendix D.2. The complete algorithm is shown in Algorithm 1.

## 4 EXPERIMENTAL SETUP

**Datasets.** We evaluate our method on both in-domain and out-of-domain datasets. For in-domain datasets, we use **Knights and Knaves (K&K)** (Xie et al., 2024) and **Musique** (Trivedi et al., 2022) for both training and evaluation. K&K is a logical reasoning dataset that requires multi-step reasoning to identify the correct characters. It consists of multiple problem difficulty levels depending on the number of characters involved. Musique is a multi-hop question answering dataset that requires retrieving and combining information from multiple sources. Both datasets naturally contain subproblems and their ground truth. We leave the exploration of dataset without intermediate ground truth for future work. For out-of-domain evaluation, we test on **GPQA** (Rein et al., 2023), **MMLU** (Hendrycks et al., 2020), and **MATH** (Hendrycks et al., 2021) to assess how well our models generalize to unseen tasks and domains. These datasets cover diverse reasoning scenarios, allowing us to comprehensively evaluate the robustness of our approach. More details about the datasets can be found in Appendix F.

**Models and Baselines.** We conduct experiments using Qwen2.5 instruct models with 1.5B and 7B parameters. To comprehensively evaluate the effectiveness of our approach, we compare it against various baselines: **Direct Inference**, where the model generates answers without explicit reasoning steps; **Chain-of-Thought (CoT)** (Wei et al., 2022), where the model performs all reasoning before generating the final answer; **SFT** (Chung et al., 2024), where the model is trained with supervised fine-tuning; **Think-answer**, where we train the same model with the standard think-answer RL methods proposed in Guo et al. (2025). We compare the baselines with two interleaved reasoning approaches: **Interleave**, our base approach without intermediate rewards; and **Interleave + IR**, our main approach with conditional *intermediate rewards* (IR) using time-discounted approach, as described in §3.2.2. For fair evaluation, we use the same setup (e.g., datasets, RL algorithms, etc.) for think-answer and interleaved training.

**Evaluation Metrics.** In this work, we use two key metrics: **Pass@1 accuracy** (How many problems are solved correctly) and **time-to-first-token (TTFT)** (How quickly the model provides answers to users). Following previous work (Meng et al., 2024; Jin et al., 2025), we use Exact Match (EM) to calculate the percentage of correct final answers against the ground truth for pass@1 score. For each test instance, we compare the model's final answer against the ground truth answer after normalization. In conventional settings, TTFT is typically measured in absolute time units (e.g., milliseconds). However, to apply it across different reasoning approaches, we define TTFT as the relative position of the first answer token in the complete response. More details on the evaluation metrics can be found in Appendix G.

**Policy Optimization.** We experiment with three policy optimization approaches: the traditional Proximal Policy Optimization (PPO) (Schulman et al., 2017) and it's two variants, Group Relative Policy Optimization (GRPO) (Shao et al., 2024) and REINFORCE++ (Hu, 2025). The key difference lies in how they estimate advantages. PPO uses a value network with Generalized Advantage Estimation (Schulman et al., 2015), while GRPO and REINFORCE++ avoid a critic network, reducing training costs. PPO is typically more stable but requires additional warm-up due to the critic, whereas GRPO and REINFORCE++ are more sample-efficient but sensitive to hyperparameter choices. We use PPO as our primary training algorithm. To ensure a fair comparison, e train models for up to 2,000 steps and report the best checkpoint for both think-answer and interleaved training. Intermediate rewards use the *Time-Discounted* method, which performed best in our experiments. Appendix B shows more details regarding training setups and stability.

## 5 RESULTS AND ANALYSIS

### 5.1 MAIN RESULTS.

Table 2 demonstrates both efficiency and accuracy benefits of interleaved reasoning. The base interleaved approach (Interleave), without using *intermediate rewards* or *intermediate ground truth*, maintains comparable Pass@1 accuracy to think-answer reasoning while reducing TTFT by more than 80% on average. This means users receive informative responses nearly *five* times sooner. The significant improvement in Pass@1 accuracy occurs when intermediate rewards are applied (Interleave + IR), leading to an average relative improvement of 12.5% across both model sizes. Moreover, training on only the datasets with intermediate ground truth, our method exhibits strong out-of-domain generalization across diverse reasoning tasks (GPQA, MMLU, and MATH), maintaining superior accuracy and reduced latency without *any* training data from that domain.

Additionally, our method also reduces overall response length by up to 37% compared to think-answer reasoning (§5.2). To validate the generated intermediate steps qualitatively, we conduct LLM-as-judge evaluation in Appendix H, which shows comparable win rates against think-answer reasoning responses. Detailed case studies in Appendix J also demonstrate how models learn to generate substantive intermediate conclusions. These findings combined indicate the effectiveness of interleaved reasoning in enhancing both model accuracy and efficiency.

Table 2: Comparison between proposed *interleaved reasoning* methods and baselines. ‡ and † represents in-domain and out-of-domain datasets, respectively. Higher Pass@1 (↑) is better, while lower TTFT (↓) is better. The best performance is bold for Pass@1, underlined for TTFT. For the non-reasoning baselines (Direct Inference, CoT, SFT) TTFT is naturally 0.

| Methods | K&K‡ | | Musique‡ | | GPQA† | | MMLU† | | MATH† | | Avg. | |
|---|---|---|---|---|---|---|---|---|---|---|---|---|
| | Pass@1↑ | TTFT↓ | Pass@1↑ | TTFT↓ | Pass@1↑ | TTFT↓ | Pass@1↑ | TTFT↓ | Pass@1↑ | TTFT↓ | Pass@1↑ | TTFT↓ |
| *Qwen2.5-1.5B-Instruct* | | | | | | | | | | | | |
| Direct Inference | 0.060 | 0.000 | 0.115 | 0.000 | 0.051 | 0.000 | 0.081 | 0.000 | 0.278 | 0.000 | 0.117 | 0.000 |
| CoT | 0.097 | 0.000 | 0.195 | 0.000 | 0.066 | 0.000 | 0.167 | 0.000 | 0.308 | 0.000 | 0.167 | 0.000 |
| SFT | 0.223 | 0.000 | 0.290 | 0.000 | 0.046 | 0.000 | 0.112 | 0.000 | 0.263 | 0.000 | 0.187 | 0.000 |
| Think-answer | 0.342 | 0.819 | 0.675 | 0.763 | 0.328 | 0.929 | 0.434 | 0.913 | **0.323** | 0.952 | 0.420 | 0.875 |
| Interleave | 0.357 | 0.118 | 0.700 | 0.210 | 0.308 | 0.181 | 0.429 | 0.189 | 0.288 | 0.163 | 0.416 | 0.172 |
| Interleave + IR | **0.533** | 0.132 | **0.710** | 0.155 | **0.489** | 0.192 | **0.460** | 0.211 | 0.313 | 0.157 | **0.501** | 0.169 |
| *Qwen2.5-7B-Instruct* | | | | | | | | | | | | |
| Direct Inference | 0.150 | 0.000 | 0.295 | 0.000 | 0.157 | 0.000 | 0.444 | 0.000 | 0.475 | 0.000 | 0.304 | 0.000 |
| CoT | 0.230 | 0.000 | 0.295 | 0.000 | 0.192 | 0.000 | 0.495 | 0.000 | 0.561 | 0.000 | 0.355 | 0.000 |
| SFT | 0.343 | 0.000 | 0.425 | 0.000 | 0.147 | 0.000 | 0.465 | 0.000 | 0.460 | 0.000 | 0.368 | 0.000 |
| Think-answer | 0.843 | 0.882 | 0.705 | 0.917 | 0.495 | 0.923 | 0.758 | 0.919 | 0.712 | 0.876 | 0.703 | 0.903 |
| Interleave | 0.803 | 0.133 | 0.735 | 0.155 | 0.505 | 0.182 | 0.769 | 0.199 | 0.707 | 0.173 | 0.704 | 0.168 |
| Interleave + IR | **0.877** | 0.129 | **0.750** | 0.167 | **0.551** | 0.166 | **0.803** | 0.178 | **0.732** | 0.167 | **0.743** | 0.161 |

Table 3: Comparison between different RL algorithms. PPO yields the best average Pass@1 as training steps increase and is more stable during training. GRPO and REINFORCE++ are sampling efficient yet less stable.

| Methods | K&K‡ | | Musique‡ | | GPQA† | | MMLU† | | MATH† | | Avg. | |
|---|---|---|---|---|---|---|---|---|---|---|---|---|
| | Pass@1↑ | TTFT↓ | Pass@1↑ | TTFT↓ | Pass@1↑ | TTFT↓ | Pass@1↑ | TTFT↓ | Pass@1↑ | TTFT↓ | Pass@1↑ | TTFT↓ |
| *GRPO* | | | | | | | | | | | | |
| Think-answer | 0.387 | 0.878 | 0.690 | 0.755 | 0.333 | 0.805 | 0.419 | 0.795 | **0.374** | 0.897 | 0.441 | 0.826 |
| Interleave | 0.383 | 0.221 | 0.650 | 0.205 | 0.409 | 0.151 | 0.424 | 0.123 | 0.313 | 0.244 | 0.436 | 0.189 |
| Interleave + IR | 0.473 | 0.164 | 0.690 | 0.132 | 0.465 | 0.133 | 0.455 | 0.230 | 0.323 | 0.198 | 0.481 | 0.171 |
| *REINFORCE++* | | | | | | | | | | | | |
| Think-answer | 0.347 | 0.859 | 0.655 | 0.794 | 0.389 | 0.868 | 0.424 | 0.912 | 0.278 | 0.751 | 0.419 | 0.837 |
| Interleave | 0.437 | 0.202 | 0.645 | 0.234 | 0.270 | 0.113 | 0.434 | 0.163 | 0.354 | 0.104 | 0.428 | 0.163 |
| Interleave + IR | 0.493 | 0.148 | **0.720** | 0.186 | 0.439 | 0.123 | 0.429 | 0.146 | 0.348 | 0.204 | 0.486 | 0.161 |
| *PPO* | | | | | | | | | | | | |
| Think-answer | 0.342 | 0.819 | 0.675 | 0.763 | 0.328 | 0.929 | 0.434 | 0.913 | 0.323 | 0.952 | 0.420 | 0.875 |
| Interleave | 0.357 | 0.118 | 0.700 | 0.210 | 0.308 | 0.181 | 0.429 | 0.189 | 0.288 | 0.163 | 0.416 | 0.172 |
| Interleave + IR | **0.533** | 0.132 | 0.710 | 0.155 | **0.489** | 0.192 | **0.460** | 0.211 | 0.313 | 0.157 | **0.501** | 0.169 |

## 5.2 ANALYSIS

In this section, we conduct a series of analyses to better understand interleaved reasoning. Unless otherwise stated, we focus on PPO using a 1.5B model with the Time-Discounted reward strategy.

**RL Algorithms Comparison.** Table 3 shows the performance differences among the three RL algorithms. PPO achieves higher Pass@1 scores for most tasks, though it generally requires more training steps to converge compared to the other two, as shown in Figure 3(b). Conversely, GRPO and REINFORCE++ demonstrate better sample efficiency, reaching competitive performance more rapidly, but they are less stable during training, which aligns with the observation from previous work (Jin et al., 2025). Overall, PPO emerges as the more stable choice for interleaved reasoning, especially when computational resources permit longer training durations, whereas GRPO and REINFORCE++ provide viable alternatives. Note that across all algorithms, our method (Interleave + IR) consistently outperforms the think-answer baseline, providing further evidence of its effectiveness.

**Reasoning Length Analysis.** Figure 3(c) shows the response length of interleaved reasoning during training. We observe that 7B and 1.5B models differ in how their response length changes. While both models achieve better performance, the response length of the 7B model grows, whereas that

Figure 3: Comparative analysis of interleaved reasoning: **(a)** Performance gap widens on harder K&K problems as difficulty increases; **(b)** Training dynamics across different RL algorithms showing convergence patterns; **(c)** Response length analysis revealing correct answers are typically shorter; **(d)** Effect of intermediate rewards on model behavior showing increased correct intermediate answers.

Table 5: Directly applying intermediate reward yields suboptimal performance. Time-discounted conditional intermediate rewards improve interleaved reasoning by incentivizing early correct steps, outperforming direct and other conditional reward methods.

| Methods | K&K[‡] | | Musique[‡] | | GPQA[†] | | MMLU[†] | | MATH[†] | | Avg. | |
|---|---|---|---|---|---|---|---|---|---|---|---|---|
| | Pass@1↑ | TTFT↓ | Pass@1↑ | TTFT↓ | Pass@1↑ | TTFT↓ | Pass@1↑ | TTFT↓ | Pass@1↑ | TTFT↓ | Pass@1↑ | TTFT↓ |
| No IR | 0.357 | 0.118 | 0.700 | 0.210 | 0.308 | 0.181 | 0.429 | 0.189 | 0.288 | 0.163 | 0.416 | 0.172 |
| Direct IR | 0.313 | 0.109 | 0.640 | 0.194 | 0.303 | 0.166 | 0.409 | 0.177 | 0.293 | 0.150 | 0.392 | 0.159 |
| Cond. IR (Partial) | 0.498 | 0.168 | 0.690 | 0.190 | 0.465 | 0.171 | 0.439 | 0.170 | 0.298 | 0.161 | 0.478 | 0.172 |
| Cond. IR (All) | 0.513 | 0.102 | 0.695 | 0.185 | 0.475 | 0.162 | 0.455 | 0.208 | 0.308 | 0.152 | 0.489 | 0.162 |
| Cond. IR (Time) | **0.533** | 0.132 | **0.710** | 0.155 | **0.489** | 0.192 | **0.460** | 0.211 | **0.313** | 0.157 | **0.501** | 0.169 |

of the 1.5B model becomes shorter. This indicates that response length is not a reliable indicator of performance, aligning with recent findings (Wang et al., 2025; Xie et al., 2025). We also observe that correct solutions are consistently shorter than incorrect ones, which suggests that failure cases often involve additional exploratory steps.

Table 4 shows the average number of tokens produced by each method between correct and incorrect responses. Beyond TTFT improvements, our approach consistently produces shorter (up to 37%) overall responses compared to think-answer reasoning. Since generation time scales linearly with token count, this directly translates to proportionally faster final answers.

Table 4: Correct answers tend to have shorter responses across all methods. Our method achieved overall shorter reasoning responses.

| Method | Correct | Incorrect | Overall |
|---|---|---|---|
| Think-Answer | 198.1 | 445.8 | 401.0 |
| Interleave+IR | 308.3 | 380.3 | 368.7 |
| Interleave | 207.6 | 259.9 | 252.8 |

We also observe that think-answer's incorrect responses are significantly longer than correct ones (more than 2x). In contrast, our interleaved approach produces shorter and more consistent lengths regardless of correctness, constraining unproductive exploration by requiring concrete intermediate conclusions. We discuss more reasoning dynamics in detail in Appendix I.

**Scaling to Harder Problems.** The K&K dataset naturally contains multiple levels of problem difficulty, with the difficulty increasing as more characters are involved. We train the model with datasets involving three, four, and five characters and evaluate on the full range of difficulties (three through eight; see Appendix F.1 for dataset details). Figure 3(a) shows that the gap between our method and the think-answer baseline widens as the difficulty increases. During logical deduction, the model builds each deduction step upon the previous one; encouraging the model to articulate and produce correct intermediate steps keeps the deductive chain intact and makes a correct final conclusion more likely. This trend indicates that interleaved reasoning not only offers practical speedups on TTFT but also improves overall reasoning, especially for harder multi-hop problems.

**Reward Strategies Comparison.** We investigate the effectiveness of different intermediate reward strategies in Table 5. Results demonstrate that directly applying intermediate rewards (Direct IR) yields lower accuracy compared to not applying intermediate reward at all (No IR). This is likely due to challenges in credit assignment inherent to reinforcement learning, where ambiguous reward signals complicate the attribution of specific actions (Leike et al., 2018). Conditional reward strategies (§3.2.2) significantly mitigate this issue by introducing intermediate rewards only when training is

Table 6: Comparison between interleaved reasoning (providing intermediate answers incrementally) versus the delayed version (providing intermediate conclusions only *after* the full reasoning trace, similar to "think-answer"). Interleaved reasoning significantly outperforms the delayed version, which suggests that timely, incremental feedback is crucial.

| Method | Use IR | K&K[‡] | | GPQA[†] | | MMLU[†] | | MATH[†] | | Avg. | |
|---|---|---|---|---|---|---|---|---|---|---|---|
| | | Pass@1↑ | TTFT↓ | Pass@1↑ | TTFT↓ | Pass@1↑ | TTFT↓ | Pass@1↑ | TTFT↓ | Pass@1↑ | TTFT↓ |
| Delayed intermediate | No | 0.287 | 0.762 | 0.273 | 0.805 | 0.409 | 0.835 | 0.298 | 0.821 | 0.317 | 0.806 |
| | Yes | 0.323 | 0.789 | 0.298 | 0.812 | 0.419 | 0.833 | 0.283 | 0.810 | 0.331 | 0.811 |
| Interleave | No | 0.357 | 0.118 | 0.308 | 0.181 | 0.429 | 0.189 | 0.288 | 0.163 | 0.346 | 0.163 |
| | Yes | **0.533** | 0.132 | **0.489** | 0.192 | **0.460** | 0.211 | **0.313** | 0.157 | **0.449** | 0.173 |

stable. The All-or-None (All) method slightly outperforms Partial Credit (Partial), suggesting that enforcing strict correctness criteria across intermediate steps better supports coherent reasoning paths than rewarding individual correct steps independently. The Time-Discounted (Time) method achieves the best performance. This result indicates that providing higher incentives for early correct reasoning steps effectively guides the model toward accurate reasoning paths.

**Impact of Intermediate Answers.** We investigate how intermediate answers influence model performance and training dynamics. First, as shown in Figure 3(d), applying intermediate rewards during training leads to a clear increase in the number of correct intermediate answers. This indicates that the reward signal effectively encourages the model to produce more accurate sub-answers, which helps steer the model along more reliable reasoning paths. Second, the timing of intermediate answers is critical. Table 6 compares our standard interleave methods with a delayed intermediate variant where intermediate answers are generated only *after* the full reasoning trace and *before* the final answer, both with and without Intermediate Rewards (IR). The delayed intermediate variant shows that generating intermediate answers *early* – not merely having them – drives both lower TTFT and higher Pass@1. Furthermore, the benefits of IR are diminished in the delayed intermediate setting, which suggests that timely, incremental feedback throughout the reasoning process is key to the effectiveness of interleaved reasoning.

**Intermediate Reward Distribution.** Figure 4 visualizes how frequently intermediate rewards are applied during training. Notably, intermediate rewards are primarily given in the early stages of training. As training progresses and the batch accuracy threshold rises, the application rate of intermediate rewards decreases. This implies that only a modest amount of intermediate reward is needed to effectively incentivize the model to produce better intermediate steps and ultimately improve final accuracy. The conditional reward strategy thus works as intended: a frequent, always-on intermediate reward is not necessary – a targeted, conditional approach is sufficient to guide the model.

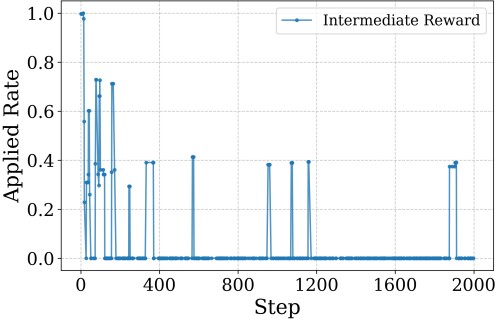

Figure 4: Visualization of intermediate reward application rate during training. The rate decreases as training progresses due to increasing batch accuracy thresholds.

## 6 CONCLUSION

We present interleaved reasoning, a novel RL paradigm that enables LLMs to alternate between reasoning and generating structured intermediate answers. Our experiments across five datasets and three RL algorithms show over 80% reduction in TTFT and a 12.5% average increase in Pass@1 accuracy. We propose a simple reward scheme that incentivizes correct intermediate steps and further enhances reasoning ability, enabling the model to generalize well to harder and unseen tasks while reducing reasoning length by up to 37%. Our comprehensive analysis provides several insights into conditional reward modeling and LLM reasoning dynamics. Interleaved reasoning offers a promising path toward more accurate, efficient, and interactive LLMs.

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

APPENDIX CONTENTS

# A    COMPARISON WITH PROCESS REWARD MODELS

Our approach differs from Process Reward Models (PRMs) in several key aspects. While PRMs typically provide token-level feedback during generation, our method evaluates the entire trajectory after completion and assigns rewards based on identifiable intermediate answers. This design choice helps avoid common PRM challenges such as reward hacking and complex training pipelines while still providing meaningful feedback on intermediate reasoning steps. Our results suggest that a simple rule-based reward can achieve similar benefits to more complex PRM implementations, in terms of guiding the model towards correct solutions.

# B    ADDITIONAL TRAINING DETAILS

**Training Setup.**    All experiments were conducted using VERL (Sheng et al., 2024), an efficient reinforcement learning framework for language models. We performed all experiments on 8 NVIDIA H100 GPUs with 80GB memory. We also used a consistent set of hyperparameters to ensure fair comparison between methods. We evaluate and save every 100 steps during training, and continue training from the last saved checkpoint if the training is interrupted (e.g., OOM). The core parameters are listed in Table 7.

Table 7: Core training hyperparameters.

| Parameter | Value |
|---|---|
| Actor learning rate | $1 \times 10^{-6}$ |
| Critic learning rate | $1 \times 10^{-6}$ |
| Train batch size | 16 |
| Validation batch size | 2048 |
| PPO mini batch size | 32 |
| PPO micro batch size | 16 |
| Critic micro batch size | 8 |
| KL coefficient | 0.001 |
| KL loss type | low variance KL |
| Max prompt length | 3096 tokens |
| Max response length | 2548 tokens |
| Sampling temperature | 0.8 |
| Number of samples per prompt | 8 |
| Stable training threshold ($\epsilon$) | 0.05 |
| Critic warmup steps | 0 |
| Evaluation frequency | 200 steps |
| Tensor model parallel size | 2 |

**Training Stability.**    To validate the stability of our reward logic and training approach, we conduct experiments using three different random seeds (0, 1, and 42) and report the results across all datasets. Table 8 presents the stability analysis results. The table shows Pass@1 accuracy and TTFT metrics for each seed across all five datasets, along with the variance ($\Delta$) calculated as the difference between maximum and minimum values across seeds. The low variance across seeds ($\Delta$ row) confirms the stability of our training approach.

Table 8: Training stability analysis across three random seeds (0, 1, 42) for the interleaved reasoning method with intermediate rewards. $\Delta$ represents the variance (max - min) across seeds, demonstrating consistent performance.

| Seed | K&K | | Musique | | GPQA | | MMLU | | MATH | | Average | |
|---|---|---|---|---|---|---|---|---|---|---|---|---|
| | Pass@1 | TTFT | Pass@1 | TTFT | Pass@1 | TTFT | Pass@1 | TTFT | Pass@1 | TTFT | Pass@1 | TTFT |
| 0 | 0.537 | 0.155 | 0.715 | 0.175 | 0.499 | 0.162 | 0.440 | 0.221 | 0.333 | 0.177 | 0.505 | 0.178 |
| 1 | 0.510 | 0.140 | 0.675 | 0.148 | 0.478 | 0.185 | 0.445 | 0.203 | 0.283 | 0.149 | 0.478 | 0.165 |
| 42 | 0.533 | 0.132 | 0.710 | 0.155 | 0.489 | 0.192 | 0.460 | 0.211 | 0.313 | 0.157 | 0.501 | 0.169 |
| $\Delta$ | 0.027 | 0.023 | 0.040 | 0.027 | 0.021 | 0.030 | 0.020 | 0.018 | 0.050 | 0.028 | 0.027 | 0.013 |

## C    THINK-ANSWER TEMPLATE

Table 9: Template for think-answer reasoning from Guo et al. (2025). prompt will be replaced with the specific reasoning question during training.

A conversation between User and Assistant. The user asks a question, and the Assistant solves it. The assistant first thinks about the reasoning process in the mind and then provides the user with the answer. The reasoning process and answer are enclosed within <think> </think> and <answer> </answer> tags, respectively, i.e., <think> reasoning process here </think> <answer> answer here </answer>. User: prompt. Assistant:

## D    ADDITIONAL REWARD DETAILS

### D.1    REWARD DEFINITION

Given the generated sequence $y$ and the ground truth answer $g = \{g_1, g_2, ..., g_N\}$, which contains all intermediate and the final answer, we perform the reward calculation based on three main components:

1. **Format Reward**: This basic component evaluates the structural aspects of the generated response. It checks whether the model properly alternates between thinking and answering phases using the designated tags (<think></think> and <answer></answer>). The reward is calculated as:

$$r_{\text{format}}(y) = \lambda_f \cdot \begin{cases} 1.0 & \text{if format is correct} \\ -1.0 & \text{if format is incorrect} \end{cases} \tag{6}$$

   where "correct" format means all tags are properly opened and closed, with proper alternation between thinking and answering. This reward is applied to both think-answer and interleaved reasoning.

2. **Final Accuracy Reward**: This component evaluates whether the final answer provided by the model matches the ground truth. We apply this reward *only when the format is correct* and use exact match for evaluation:

$$r_{\text{final}}(x, y) = \lambda_a \cdot \begin{cases} 2.0 & \text{if } y_{\text{answer}}^{(N)} = g_N \\ -1.5 & \text{if } y_{\text{answer}}^{(N)} \neq g_N \\ -2.0 & \text{if answer is not parseable} \end{cases} \tag{7}$$

   where $g_N$ is the final ground truth answer. For structured outputs (like numerical answers or multi-choice questions), we normalize both the model's answer and ground truth and use exact match for evaluation. This reward is applied to both think-answer and interleaved reasoning.

3. **Intermediate Accuracy Reward**: This component provides rewards for correct intermediate answers, calculated using one of the three strategies discussed in Section 3.2.2. The intermediate reward is applied conditionally, as detailed in Algorithm 1, and is only used for interleaved reasoning.

### D.2    CONDITIONAL INTERMEDIATE REWARD

We provide detailed descriptions on three intermediate reward strategies in this section. The base intermediate reward value $R_{\text{base}}$ is set to be 0.5 in this work.

1. **All-or-None**: This strategy requires all intermediate answers to be correct in sequence to receive any reward. The reward calculation is:

$$r_{\text{intermediate}}^{\text{all-or-none}}(x, y) = \begin{cases} R_{\text{base}} & \text{if } \text{Correct}(y_{\text{answer}}^{(k)}), \ \forall k \in [1, N-1], \\ 0 & \text{otherwise} \end{cases} \tag{8}$$

   This strategy is the most demanding but ensures the model maintains a consistent reasoning path throughout.

2. **Partial Credit**: This strategy rewards each correct intermediate answer independently, providing partial credit regardless of other steps:

$$r_{\text{intermediate}}^{\text{partial}}(x, y) = \frac{R_{\text{base}}}{N-1} \sum_{k=1}^{N-1} \text{Correct}(y_{\text{answer}}^{(k)}) \tag{9}$$

This approach is more forgiving, allowing the model to recover from early mistakes while still incentivizing correct intermediate steps.

3. **Time-Discounted**: This strategy awards the full base reward $R_{\text{base}}$ when every intermediate answer is correct. If any intermediate answers are missing or wrong, the reward is shared among the correct ones with higher weight on earlier appears correct answers. Formally,

$$r_{\text{intermediate}}^{\text{time-disc}}(x, y) = \begin{cases} R_{\text{base}}, & \text{if } S_{\text{correct}} = g, \\ R_{\text{base}} \dfrac{1}{|g|} \displaystyle\sum_{g_j \in S_{\text{correct}}} \dfrac{1}{k_j}, & \text{otherwise}, \end{cases} \tag{10}$$

where $S_{\text{correct}} \subseteq g$ is the set of ground-truth intermediate answers that the model outputs at least once, $k_j$ is the index of the first step in which the model's answer matches $g_j$, and $|g|$ is the total number of ground-truth intermediate answers. The harmonic weight $1/k_j$ gives greater credit to earlier correct answers while still granting some credit to later ones. Note that the time-discounted partial reward calculation will *not* be used if all intermediate answers are correct. Therefore the model receives a larger reward when all intermediate answers are correct, and the reward quickly drops even if one intermediate answer is incorrect. This design choice was intentionally made to strongly incentivize the model to generate all correct intermediate steps, rather than being satisfied with partial correctness.

Our analysis reveals that early correct token generation actually enhances reasoning (§5.1). Our reward strategy analysis (Table 5) shows that the time-aware reward is more effective than the time-agnostic reward, suggesting that generating correct intermediate answers early helps the model reason more efficiently. Consequently, *behaviors such as backtracking or rethinking are less frequently observed*, as the reward explicitly encourages the model to generate early intermediate answers. While our analysis demonstrates that early correct token generation is beneficial, investigating the depth and breadth of interleaved reasoning represents an interesting direction for future research.

## E    BEYOND INTERMEDIATE GROUND TRUTH

Intermediate ground truths help during training, but they are not a strict requirement for deploying interleaved reasoning. *Firstly*, strong generalization mitigates training requirements. Although our training uses datasets with intermediate ground truths (K&K and Musique), the resulting models generalize to tasks that do not provide any intermediate annotations at evaluation time (Table 2). Trained only on K&K and Musique, our models achieve superior performance on MATH, GPQA, and MMLU – none of which include intermediate ground truths for evaluation. This reduces the practical limitation of needing intermediate annotations across domains.

*Secondly*, alternative supervision methods are readily available. The number of concurrent works using the model's internal confidence as reward signals (Li et al., 2025a; Agarwal et al., 2025) or process rewards (Khalifa et al., 2025; Zhang et al., 2025) provides viable approaches to apply our method to datasets lacking explicit intermediate annotations. While we do not explore these combinations in this work, existing techniques for generating intermediate supervision can be integrated with our method to expand its applicability, representing an exciting future direction.

*Lastly*, we conduct additional ablations on GSM8K with Qwen2.5-1.5B model (Table 10), without intermediate ground truths. Even *without* intermediate rewards, interleaving preserves accuracy while substantially reducing TTFT. The results indicate that the interleaving structure itself brings substantial responsiveness gains by-default with comparable accuracy to standard think-answer training, even when intermediate rewards are absent.

**Algorithm 1** Intermediate Reward Calculation

---

1: **Input:** Generated sequence $y$, ground truth intermediate answers $g = \{g_1, g_2, ..., g_N\}$, current training batch $B$, reward strategy $S$
2: **Parameters:** Base reward value $R_{\text{base}}$, stable training threshold $\epsilon$
3: **Output:** Intermediate reward value
4: Parse $y$ to extract all intermediate answers $y_{\text{answer}} = \{y_{\text{answer}}^{(1)}, \dots, y_{\text{answer}}^{(N)}\}$, where $y_{\text{answer}}^{(N)}$ is the final answer
5: is_final_correct $\leftarrow$ Correct$(y_{\text{answer}}^{(N)})$
6: is_format_valid $\leftarrow$ Format$(y)$
7: is_progressing $\leftarrow (\text{Acc}(B) > \text{Acc}(B-1) - \epsilon)$
8: **if** is_final_correct AND is_format_valid AND is_progressing **then**
9:     reward_sum $\leftarrow 0$
10:     **if** $S = $ "All-or-None" **then**
11:         all_correct $\leftarrow$ TRUE
12:         **for** $k = 1$ to $N-1$ **do**
13:             **if** NOT Correct$(y_{\text{answer}}^{(k)})$ **then**
14:                 all_correct $\leftarrow$ FALSE
15:                 **break**
16:             **end if**
17:         **end for**
18:         **if** all_correct **then**
19:             reward_sum $\leftarrow R_{\text{base}}$
20:         **end if**
21:     **else if** $S = $ "Partial Credit" **then**
22:         **for** $k = 1$ to $N-1$ **do**
23:             **if** Correct$(y_{\text{answer}}^{(k)})$ **then**
24:                 reward_sum $\leftarrow$ reward_sum $+ R_{\text{base}}/N$
25:             **end if**
26:         **end for**
27:     **else if** $S = $ "Time-Discounted" **then**
28:         correct_step $\leftarrow \{\}$ {Track all correct steps}
29:         **for** $k = 1$ to $N-1$ **do**
30:             **for** each required answer $g_j$ in $g$ **do**
31:                 **if** $g_j$ not in correct_step AND Correct$(y_{\text{answer}}^{(k)})$ **then**
32:                     correct_step$[g_j] \leftarrow k$
33:                 **end if**
34:             **end for**
35:         **end for**
36:         **if** $|\text{correct\_step}| = |g|$ **then**
37:             reward_sum $\leftarrow R_{\text{base}}$
38:         **else**
39:             sum_weights $\leftarrow \sum_{step \in correct\_step} 1/step$
40:             reward_sum $\leftarrow (\text{sum\_weights}/|g|) \cdot R_{\text{base}}$
41:         **end if**
42:     **end if**
43:     **return** reward_sum
44: **else**
45:     **return** $0$
46: **end if**

---

Table 10: Our approach provides efficiency benefits even for datasets without intermediate ground truths, with comparable accuracy gains to the think-answer structure.

| Algo. | Method | Acc. | TTFT |
|-------|--------|------|------|
| PPO | Normal | 79.3 | 88.8 |
|  | Interleaved | **79.7** | 26.2 |
| GRPO | Normal | 78.4 | 88.6 |
|  | Interleaved | **78.6** | 26.2 |
| RF++ | Normal | **78.1** | 88.5 |
|  | Interleaved | **78.1** | 25.9 |

## F    DATASET DETAILS

### F.1    IN-DOMAIN

**Knights and Knaves (K&K).**    K&K is a logical reasoning dataset that requires multi-step reasoning to identify the correct characters (Xie et al., 2024). The dataset contains problems with varying difficulty levels based on the number of characters involved. In our experiments, we use problems with 3, 4, and 5 characters for both training and evaluation. Each difficulty level consists of 900 training examples and 100 test examples. To evaluate generalization across difficulty levels, we also test our models on problems with 6, 7, and 8 characters, which were not seen during training (Figure 3(a)). Our results indicate that interleaved reasoning is particularly effective for more challenging problems.

**Musique.**    Musique is a multi-hop question answering dataset that requires retrieving and combining information from multiple sources (Trivedi et al., 2022). Problems in Musique are categorized by the number of reasoning hops needed (i.e., 2-hop, 3-hop, 4-hop). For our experiments, we use 3-hop and 4-hop questions, with 900 training examples and 100 test examples for each hop category. For efficient training and inference, we select only up to 1,000 tokens in total for the context, which includes all the supporting documents and a portion of distraction documents. Both K&K and Musique naturally contain intermediate reasoning steps and ground truth, making them ideal for training and evaluating interleaved reasoning approaches.

### F.2    OUT-OF-DOMAIN

**GPQA.**    We use the GPQA-diamond version (Rein et al., 2023), which consists of 198 data points. GPQA is crafted by domain experts in biology, physics, and chemistry, designed to assess LLMs advanced reasoning and knowledge.

**MMLU.**    We use MMLU-redux-2.0 (Gema et al., 2024), a cleaned and reannotated version of MMLU (Hendrycks et al., 2020). To match with GPQA, we select a subset of 198 data points from domains requiring formal reasoning: college computer science, college mathematics, abstract algebra, formal logic, college physics, and machine learning.

**MATH.**    We also use 198 data points from the level 5 subset of MATH (Hendrycks et al., 2021), which are the most challenging problems within the dataset. These problems require complex mathematical reasoning and often involve multiple steps of computation and logical deduction.

## G    EVALUATION METRICS

**Pass@1 Accuracy.**    Pass@1 accuracy measures the proportion of problems that the model solves correctly on its first attempt. We follow the evaluation methodology established in prior work (Wei et al., 2022; Guo et al., 2025; Jin et al., 2025), using Exact Match (EM) to determine correctness. For each test instance, we compare the model's final answer against the ground truth answer after normalizing both (removing punctuation, converting to lowercase, and standardizing numerical formats). A prediction is considered correct only if it exactly matches the normalized ground truth.

**Time-to-First-Token (TTFT).** TTFT measures how quickly a model produces its first useful output to the user. While traditional approaches measure TTFT in absolute time (milliseconds), we normalize TTFT as the ratio of the first answer token's position to the total response length to ensure fair comparison across different model configurations and reasoning strategies:

$$\text{TTFT} = \frac{\text{Position of first answer token}}{\text{Total response length}} \tag{11}$$

This normalized metric ranges from 0 to 1, where lower values indicate faster initial responses. This metric is particularly important for interactive applications where immediate response could vastly improve user experience.

### G.1 Substring Exact Match (SubEM) and Reward Hacking

We initially experimented with SubEM as an additional evaluation metric for intermediate answers. SubEM is more lenient than EM – it measures whether the ground truth answer appears as a *substring* in the model's response. We found that models trained with SubEM quickly learned to generate *excessively long* intermediate answers containing numerous potential responses, significantly increasing the probability of including the correct answer somewhere in the text. For example, instead of generating a concise intermediate step "The value is 42," models would produce verbose outputs like "Let me consider different possibilities: the value is 41, the value is 42, the value is 43 ..." This gaming behavior provided no pedagogical value and undermined the training.

This observation aligns with prior findings in reinforcement learning, where models exploit evaluation metrics in unintended ways (Xie et al., 2025), which is as known as reward hacking. Therefore, we use EM as our main evaluation metric.

## H Qualitative Analysis of Interleaved Reasoning

To complement our quantitative findings on significant time-to-first-token (TTFT) reduction, we conduct a qualitative evaluation using an LLM-based judge (`gpt-4o-mini-2024-07-18`) to assess the value of interleave reasoning. Specifically, we compared two versions of the interleaved method (with and without intermediate rewards) against the standard think-answer method. For each problem that are solved correctly by all three methods (126 problems in total, 38 in-domain, 88 out-of-domain), we presented the problem statement and the model responses to the LLM evaluator, asking it to rate each answer on three criteria: (1) clarity and usefulness of intermediate steps, (2) timeliness and informativeness of feedback, and (3) overall user experience. The LLM was instructed to mimic a human evaluator and assign scores for each criterion and to select a winner between the two methods for each example. The evaluation prompt is shown in Appendix H.1.

We calculate the win rates for each method, as shown in Table 11. Win rate is calculated as the percentage of pairwise wins (excluding ties). The results show that the base interleaved method (without intermediate rewards) had a lower win rate compared to think-answer, indicating that not all intermediate answers were useful by default. However, when intermediate rewards were used to encourage the model to produce more meaningful intermediate answers, the interleaved method outperformed think-answer in terms of both win rate and qualitative scores, highlighting the importance of intermediate rewards in enhancing the user experience.

Table 11: LLM-based qualitative evaluation: average win rates and average scores by domain.

| Dataset Group | Think-Ans vs. Interleave | | Think-Ans vs. Inter+IR | |
|---|---|---|---|---|
| | Think-Ans Win (%) | Inter Win (%) | Think-Ans Win (%) | Interleave+IR Win (%) |
| In-domain | 36.7 | **63.4** | 43.4 | **56.7** |
| Out-of-domain | **70.1** | 29.9 | **52.1** | 47.9 |
| *Average* | **53.4** | 46.7 | 48.6 | **51.4** |

## H.1 LLM-JUDGE EVALUATION PROMPT

The following prompt was used to instruct the LLM judge for qualitative evaluation:

---

**Evaluation Prompt**

You are an expert evaluator of large language model reasoning. You are given a multi-hop problem and two model-generated answers. The first answer uses interleaved reasoning: it alternates between thinking and answering, providing intermediate answers as soon as they are derived. The second answer uses the traditional think-answer reasoning: it completes all reasoning before providing the final answer. For each answer, your task is to rate it on a scale from 1 (very poor) to 10 (excellent) for each of the following criteria:

- Clarity and usefulness of intermediate reasoning steps
- Timeliness and informativeness of feedback (does the response help the user understand the reasoning?)
- Overall user experience

**Instructions:**

- Assign a score (1-10) for each criterion for both answers.
- After scoring, briefly explain your reasoning for the scores.
- Respond in JSON as:

```
{
  "interleave": {
    "clarity_usefulness": <int>,
    "timeliness_informativeness": <int>,
    "overall_experience": <int>
  },
  "think_answer": {
    "clarity_usefulness": <int>,
    "timeliness_informativeness": <int>,
    "overall_experience": <int>
  },
  "explanation": "<your reasoning for these scores>"
}
```

**Problem:**
`<problem_text>`

**Interleaved Reasoning Answer:**
`<interleave_response>`

**Think-Answer Reasoning Answer:**
`<think_answer_response>`

---

## I BACKTRACKING IN INTERLEAVED REASONING

The interleaved reasoning structure means that intermediate answers are committal, which restricts backtracking seen in regular CoT reasoning. However, while backtracking might potentially improve reasoning, it could also lead to overthinking and inefficiency (Li et al., 2025c). Our results suggest that interleaved reasoning encourages more efficient reasoning by incentivizing models to reach correct intermediate conclusions early (§5.1), reducing the need for backtracking. Our results also shows that the structural benefits of interleaved reasoning outweigh potential committal disadvantages, achieving higher accuracy, lower TTFT (Table 2), and shorter reasoning traces (Table 4). We leave the exploration of backtracking in interleaved reasoning for future work.

## J CASE STUDIES OF INTERLEAVED REASONING

We present case studies to gain a deeper understanding of interleaved reasoning. We focus on Qwen2.5-1.5B-Instruct that was trained using PPO and the Time-Discounted reward strategy. We present the examples from both in-domain (K&K, Musique) and out-of-domain (MATH, MMLU, GPQA) datasets. The model successfully generalizes to unseen, complex problems, demonstrating its ability to break down problems into logical steps, provide intermediate conclusions, and arrive at final answers. Green for correct answers and red for incorrect answers.

---

**MATH**

*The model successfully decomposes the problem into smaller steps and correctly computes the final answer.*

---

**Question**:
Expand the product $(x - 2)^2(x + 2)^2$. What is the product of the nonzero coefficients of the resulting expression, including the constant term?

**Ground Truth**: -128

**Interleaved Reasoning**:
<think>First, let's expand the product step by step. We start with the expression $(x - 2)^2(x + 2)^2$. We can simplify this by recognizing that $(x - 2)(x + 2) = x^2 - 4$. So, the expression becomes $(x^2 - 4)^2$.</think>
<answer>$(x^2 - 4)^2$</answer>
<think>Next, we need to expand $(x^2 - 4)^2$. This is a square of a binomial, which can be expanded using the formula $(a - b)^2 = a^2 - 2ab + b^2$.</think>
<answer>$(x^2 - 4)^2 = x^4 - 8x^2 + 16$</answer>
<think>The expanded form of the expression is $x^4 - 8x^2 + 16$. The nonzero coefficients are 1, -8, and 16. We need to find the product of these coefficients.</think>
<answer>The product of the coefficients 1, -8, and 16 is $1 \times -8 \times 16 = -128$.</answer>
<think>The final answer is:</think>
<answer>-128</answer>

---

**MATH**

*A failed case: the model follows the interleaved structure but makes a calculation error in step 4, resulting in an incorrect final answer. However, with interleaved reasoning, the incorrect intermediate answer can potentially be spotted early by users.*

---

**Question**:
Let $f(x)$ be the polynomial

$$f(x) = 3x^4 + 5x^2 - 9x - 2.$$

If $g(x)$ is equal to the polynomial $f(x - 1)$, what is the sum of the coefficients of $g$?

**Ground Truth**: -2

**Interleaved Reasoning**:
<think> First, we need to find the polynomial $g(x)$ by substituting $x - 1$ into $f(x)$. </think>
<answer> $f(x) = 3x^4 + 5x^2 - 9x - 2$ </answer>
<think> Second, we substitute $x - 1$ into $f(x)$ to get $g(x)$. </think>
<answer> $g(x) = 3(x - 1)^4 + 5(x - 1)^2 - 9(x - 1) - 2$ </answer>
<think> Third, we expand $g(x)$ to find the polynomial in standard form. </think>
<answer> $g(x) = 3(x^4 - 4x^3 + 6x^2 - 4x + 1) + 5(x^2 - 2x + 1) - 9(x - 1) - 2$ </answer>
<think> Fourth, we simplify $g(x)$ to find the coefficients. </think>
<answer> $g(x) = 3x^4 - 12x^3 + 24x^2 - 24x + 3 + 5x^2 - 10x + 5 - 9x + 9 - 2$ </answer>
<think> Fifth, we combine like terms to get the polynomial in standard form. </think>
<answer> $g(x) = 3x^4 - 12x^3 + 29x^2 - 43x + 15$ </answer>
<think> Sixth, we find the sum of the coefficients of $g(x)$. </think>
<answer> $3 - 12 + 29 - 43 + 15 = -4$ </answer>

```
<think> Seventh, the final answer is:</think>
<answer> -4 </answer>
```

### Knights and Knaves

*The model successfully breaks down a multi-hop logic puzzle, providing correct intermediate conclusions at each step, and arrives at the correct final answer.*

**Question**:
A very special island is inhabited only by knights and knaves. Knights always tell the truth, and knaves always lie. You meet 5 inhabitants: Victoria, Mia, Sebastian, Ethan, and Olivia. "Mia is a knave and Ethan is a knave" - Victoria.
Mia said that If Mia is a knight then Sebastian is a knave.
Sebastian expressed that Victoria is a knave if and only if Ethan is a knave.
Ethan remarked, "Sebastian is a knight or Victoria is a knave".
"Sebastian is a knight and Sebastian is a knave" - Olivia.
So who is a knight and who is a knave?
**Ground Truth**: (1) Victoria is a knave (2) Mia is a knave (3) Sebastian is a knight (4) Ethan is a knight (5) Olivia is a knave
**Interleaved Reasoning**:
`<think>` First, consider Victoria's statement: 'Mia is a knave and Ethan is a knave'. This must be a lie because an implication can only be true if both conditions are false. `</think>`
`<answer>` Victoria is a knave `</answer>`
`<think>` Second, consider Sebastian's statement: 'Victoria is a knave if and only if Ethan is a knave'. Since Victoria is a knave (from our previous deductions) and Sebastian is a knight (since he always lies), Sebastian must be a knight. `</think>`
`<answer>` Sebastian is a knight `</answer>`
`<think>` Third, consider Mia's statement: 'If Mia is a knight then Sebastian is a knave'. Since Mia is a knight (since she always lies) and Sebastian is a knight (since he always lies), Mia must be a knave. `</think>`
`<answer>` Mia is a knave `</answer>`
`<think>` Fourth, consider Ethan's statement: 'Sebastian is a knight or Victoria is a knave'. Since Sebastian is a knight (since he always lies) and Victoria is a knave (since she always lies), Ethan must be a knight. `</think>`
`<answer>` Ethan is a knight `</answer>`
`<think>` Fifth, consider Olivia's statement: 'Sebastian is a knight and Sebastian is a knave'. This must be a lie because an implication can only be true if both conditions are simultaneously true, which is impossible. `</think>`
`<answer>` Olivia is a knave `</answer>`
`<think>` So, the final answer is:`</think>`
`<answer>`(1) Victoria is a knave (2) Mia is a knave (3) Sebastian is a knight (4) Ethan is a knight (5) Olivia is a knave`</answer>`

### Musique

*The model sequentially solves each sub-question, yielding the correct final answer. Occasionally, the model provides intermediate answers that are duplicates of the final answer, which is not ideal.*

**Question**:
Where does Plymouth rank in population among the cities of the country having a gallery displaying the painting named The Woman Taken in the main subject of milkman joke?
**Ground Truth**: 30th
**Interleaved Reasoning**:
`<think>` First, identify the country where Plymouth is located. `</think>`
`<answer>` United Kingdom `</answer>`
`<think>` Second, identify the painting named The Woman Taken in Adultery. `</think>`

```
<answer> The Woman Taken in Adultery </answer>
<think> Third, identify the gallery displaying this painting. </think>
<answer> National Gallery </answer>
<think> Fourth, determine Plymouth's ranking in population among the cities of the United
Kingdom. </think>
<answer> 30th </answer>
<think> Fifth, state the final answer. </think>
<answer> 30th </answer>
```

## MMLU

*The model translates logic formulas stepwise, mapping each part to natural language and
selecting the correct answer.*

**Question**:
Use the following key to translate the given formula of PL to natural, English sentences.
A: Marina reads a Percy Jackson book.
B: Izzy plays Minecraft.
C: Emily stops working.
D: Russell makes dinner.
E: Ashleigh stops by.
$\sim (A \supset B) \cdot (B \supset \sim E)$
A) It's not the case that Marina's reading a Percy Jackson book entails that Izzy plays
Minecraft, but Izzy's playing Minecraft does entail that Ashleigh doesn't stop by.
B) If Marina doesn't read a Percy Jackson book, then Izzy plays Minecraft, which entails that
Ashleigh doesn't stop by.
C) Marina's reading a Percy Jackson book does not entail that Izzy plays Minecraft, but Izzy
plays Minecraft provided that Ashleigh doesn't stop by.
D) It's not true that Marina reads a Percy Jackson book only when Izzy plays Minecraft, but
Izzy plays Minecraft only when Ashleigh stops by.
**Ground Truth**: A
**Interleaved Reasoning**:
<think>First, identify the logical structure of the given formula. The formula is a conjunc-
tion of two main parts: $\sim (A \supset B)$ and $(B \supset \sim E)$. The symbol $\supset$ represents material
implication ("entails", "if...then...", "only when"), $\sim$ represents negation ("not", "it's not the
case that"), and $\cdot$ represents conjunction ("and", "but").</think>
<answer>The formula is $\sim (A \supset B) \cdot (B \supset \sim E)$.</answer>
<think>Second, translate the first part of the conjunction: $\sim (A \supset B)$. $A \supset B$ translates
to "Marina reads a Percy Jackson book entails Izzy plays Minecraft" or "If Marina reads a
Percy Jackson book, then Izzy plays Minecraft." The negation $\sim (A \supset B)$ means "It is not
the case that (Marina reads a Percy Jackson book entails Izzy plays Minecraft)."</think>
<answer>$\sim (A \supset B)$ translates to "It's not the case that Marina's reading a Percy Jackson
book entails that Izzy plays Minecraft."</answer>
<think>Third, translate the second part of the conjunction: $(B \supset \sim E)$. $B$ translates to
"Izzy plays Minecraft." $\sim E$ translates to "Ashleigh doesn't stop by." So, $B \supset \sim E$ translates
to "Izzy plays Minecraft entails that Ashleigh doesn't stop by" or "If Izzy plays Minecraft,
then Ashleigh doesn't stop by."</think>
<answer>$(B \supset \sim E)$ translates to "Izzy's playing Minecraft does entail that Ashleigh
doesn't stop by."</answer>
<think>Fourth, combine the translations of the two parts with a conjunction (e.g., "but"
or "and"). The overall formula $\sim (A \supset B) \cdot (B \supset \sim E)$ combines these two state-
ments.</think>
<answer>Combining the parts: "It's not the case that Marina's reading a Percy Jackson
book entails that Izzy plays Minecraft, but Izzy's playing Minecraft does entail that Ashleigh
doesn't stop by."</answer>
<think>Fifth, compare this translation with the given options to select the correct one.
Option A matches our translation. Option B incorrectly translates $\sim (A \supset B)$ as "If Marina

doesn't read a Percy Jackson book, then Izzy plays Minecraft". Option C incorrectly translates $(B \supset \sim E)$ as "Izzy plays Minecraft provided that Ashleigh doesn't stop by", which would be $(\sim E \supset B)$. Option D incorrectly translates $(B \supset \sim E)$ as "Izzy plays Minecraft only when Ashleigh stops by", which would be $(B \supset E)$. Therefore, option A is the correct translation.</think>
<answer>The final answer is: A</answer>

## MMLU

*The model successfully applies interleaved reasoning to derive and combine gradients step by step, leading to the correct answer.*

**Question**:
Suppose we have the following objective function: $\arg\min_w \frac{1}{2}\|Xw - y\|_2^2 + \frac{1}{2}\gamma\|w\|_2^2$. What is the gradient of $\frac{1}{2}\|Xw - y\|_2^2 + \frac{1}{2}\lambda\|w\|_2^2$ with respect to $w$?
A) $\nabla_w f(w) = (X^T X + \lambda I)w - X^T y + \lambda w$
B) $\nabla_w f(w) = X^T Xw - X^T y + \lambda$
C) $\nabla_w f(w) = X^T Xw - X^T y + \lambda w$
D) $\nabla_w f(w) = X^T Xw - X^T y + (\lambda + 1)w$
**Ground Truth**: A
**Interleaved Reasoning**:
<think>First, let's identify the objective function and the terms involved. The objective function is given by:</think>
<answer>the objective function is $\frac{1}{2}\|Xw - y\|_2^2 + \frac{1}{2}\gamma\|w\|_2^2$</answer>
<think>Now, let's break down the terms:</think>
<answer>the terms are $\frac{1}{2}\|Xw - y\|_2^2$ and $\frac{1}{2}\gamma\|w\|_2^2$</answer>
<think>Next, we need to find the gradient of the first term $\frac{1}{2}\|Xw - y\|_2^2$ with respect to $w$. Recall that the gradient of $\|z\|_2^2$ with respect to $z$ is $2z$. So, the gradient of $\frac{1}{2}\|Xw - y\|_2^2$ is:</think>
<answer>the gradient of $\frac{1}{2}\|Xw - y\|_2^2$ is $X^T(Xw - y)$</answer>
<think>Now, let's find the gradient of the second term $\frac{1}{2}\gamma\|w\|_2^2$ with respect to $w$. The gradient of $\|w\|_2^2$ with respect to $w$ is $2w$. So, the gradient of $\frac{1}{2}\gamma\|w\|_2^2$ is:</think>
<answer>the gradient of $\frac{1}{2}\gamma\|w\|_2^2$ is $\gamma w$</answer>
<think>Now, we can combine these gradients to find the gradient of the entire objective function:</think>
<answer>the gradient of the entire objective function is $X^T(Xw - y) + \gamma w$</answer>
<think>Now, let's compare this with the given options:</think>
<answer>The correct option is A) $\nabla_w f(w) = (X^T X + \lambda I)w - X^T y + \lambda w$</answer>

## GPQA

*The model answers the problem step by step, ruling out distractors and resulting in the correct final answer.*

**Question**:
In the lab, a chemist discovers an unnamed bottle with a white solid inside. He gathers the following spectral information in order to identify the liquid. Make a reasonable structural suggestion for the unidentified drug using the information provided.
Mass: molecular ion peak at m/z = 156 (100%)
a peak at m/z = 158 (32%)
IR : broad peak from 3500-2700 $cm^{-1}$, strong sharp peak at 1720 $cm^{-1}$
$^1$H NMR: 11.0 ppm (s, 1H), 8.02 ppm (d, 2H), 7.72 (d, 2H)
A) Phenyl chloroformate
B) 2-chlorobenzoic acid
C) 3-Chloro-2-hydroxybenzaldehyde
D) 4-chlorobenzoic acid

**Ground Truth**: D

**Interleaved Reasoning**:

<think>First, analyze the Mass Spectrum data. A molecular ion peak ($M^+$) at m/z = 156 and an M+2 peak at m/z = 158 with an intensity ratio of approximately 100:32 (or 3:1) is characteristic of a compound containing one chlorine atom ($^{35}Cl/^{37}Cl$ isotopic abundance). The molecular weight of $C_7H_5ClO_2$ is approximately 156.5 g/mol (using $^{35}Cl$), which matches the $M^+$ peak. All given options have this molecular formula.</think>

<answer>The mass spectrum indicates a molecular weight of approximately 156 and the presence of one chlorine atom. All options (Phenyl chloroformate, 2-chlorobenzoic acid, 3-Chloro-2-hydroxybenzaldehyde, 4-chlorobenzoic acid) have the molecular formula $C_7H_5ClO_2$.</answer>

<think>Second, analyze the IR Spectrum. A very broad peak spanning 3500-2700 $cm^{-1}$ is highly characteristic of the O-H stretch of a carboxylic acid, which is broadened due to hydrogen bonding. The strong, sharp peak at 1720 $cm^{-1}$ is characteristic of a C=O (carbonyl) stretch, consistent with a carboxylic acid (typically 1700-1725 $cm^{-1}$). This strongly suggests the presence of a -COOH group.</think>

<answer>The IR spectrum strongly suggests the presence of a carboxylic acid group (-COOH) due to the broad O-H stretch (3500-2700 $cm^{-1}$) and the C=O stretch (1720 $cm^{-1}$).</answer>

<think>Third, analyze the $^1$H NMR Spectrum. The signal at 11.0 ppm (singlet, 1H) is highly deshielded and is characteristic of a carboxylic acid proton (-COOH). The signals at 8.02 ppm (doublet, 2H) and 7.72 ppm (doublet, 2H) are in the aromatic region. A pattern of two doublets, each integrating to 2 protons, is indicative of a para-substituted benzene ring. The two protons of one type are chemically equivalent, and the two protons of the other type are also chemically equivalent, and they are ortho to each other respectively.</think>

<answer>The $^1$H NMR spectrum shows a carboxylic acid proton at 11.0 ppm and a para-substituted benzene ring pattern (two doublets, each 2H, at 8.02 ppm and 7.72 ppm).</answer>

<think>Fourth, evaluate the options based on the combined spectral data: A) Phenyl chloroformate: Contains an ester-like C=O and an acid chloride C-Cl. IR for C=O in R-O-CO-Cl is typically higher (around 1760-1780 $cm^{-1}$). It lacks the carboxylic acid O-H and the acidic proton at 11.0 ppm. B) 2-chlorobenzoic acid: Is a carboxylic acid, consistent with IR and the 11.0 ppm NMR signal. However, an ortho-substituted benzene ring (1,2-disubstituted) would typically give a more complex pattern for the 4 aromatic protons (e.g., 4 distinct signals or complex multiplets), not two doublets of 2H each. C) 3-Chloro-2-hydroxybenzaldehyde: Contains an aldehyde and a phenolic -OH. An aldehyde C=O IR is plausible ( 1720-1740 $cm^{-1}$), but the broad O-H from 3500-2700 $cm^{-1}$ points to a carboxylic acid, not typically a phenol (phenols are usually sharper, 3200-3600 $cm^{-1}$). The aldehyde proton (-CHO) signal is typically around 9-10 ppm. The aromatic region for a 1,2,3-trisubstituted benzene would show 3 distinct aromatic protons. D) 4-chlorobenzoic acid: Is a carboxylic acid. This fits the IR data (broad O-H, C=O at 1720 $cm^{-1}$) and the NMR signal at 11.0 ppm (-COOH). It is a para-substituted benzene ring (chlorine at position 4, carboxylic acid at position 1). This substitution pattern gives rise to two sets of equivalent aromatic protons (H-2/H-6 and H-3/H-5), which would appear as two doublets, each integrating to 2H. This perfectly matches the NMR signals at 8.02 ppm and 7.72 ppm.</think>

<answer>Comparing the spectral interpretations with the options: A) Phenyl chloroformate is inconsistent with IR (no broad O-H) and NMR (no signal at 11 ppm). B) 2-chlorobenzoic acid is inconsistent with the aromatic NMR pattern (would not be two doublets of 2H each). C) 3-Chloro-2-hydroxybenzaldehyde is inconsistent with IR (carboxylic acid O-H unlikely) and NMR (aldehyde H at 9-10 ppm, different aromatic pattern). D) 4-chlorobenzoic acid is fully consistent with all spectral data: MS indicates $C_7H_5ClO_2$, IR indicates a carboxylic acid, and $^1$H NMR indicates a carboxylic acid proton and a para-substituted benzene ring.</answer>

<think>The final answer is therefore 4-chlorobenzoic acid.</think>

<answer>D) 4-chlorobenzoic acid</answer>

---

**GPQA**

*A failed case: the model occasionally fails to follow the interleaved structure. The model also sometimes generates shorter thinking and longer answers than necessary.*

---

**Question**:
Reaction of solid A with 8 equivalents of gas B forms bright red product C.
When C reacts with 2 equivalents of gas D, it produces the extremely hazardous product E.
C reacts with water to reform A plus two different acids F and G. F is a strong acid while G is a weak acid.
D reacts with B in a 1:1 ratio to form H, which is used as a solvent. What is the molecular symmetry group of E?
A) $C_{2v}$
B) $D_{4h}$
C) $D_{\infty h}$
D) $C_2$

**Ground Truth**: A

**Interleaved Reasoning**:
<think>First, let's identify the products and their properties:</think>
<answer>The products are C, E, F, and G. C is a bright red product, E is an extremely hazardous product, F is a strong acid, and G is a weak acid. D reacts with B in a 1:1 ratio to form H, which is used as a solvent.</answer>
<think>Next, let's analyze the reaction of C with water to reform A plus two different acids F and G:</think>
<answer>The reaction of C with water to reform A plus two different acids F and G is a typical acid-base reaction. Since C is a bright red product, it is likely a metal oxide or a metal hydroxide. When it reacts with water, it will form a metal hydroxide and a metal ion, which will then react with the acid to form the two different acids F and G.</answer>
<think>Now, let's consider the reaction of E with D to produce H:</think>
<answer>The reaction of E with D to produce H is a typical acid-base reaction. Since E is an extremely hazardous product, it is likely a strong acid. When it reacts with D, it will form H, which is used as a solvent.</answer>
<think>Finally, let's determine the molecular symmetry group of E:</think>
<answer>The molecular symmetry group of E is $D_{\infty h}$. This is because E is an extremely hazardous product, which suggests that it has a high degree of symmetry. The $D_{\infty h}$ group has 16 symmetry operations, including 8 mirror planes and 8 axes of inversion. This group is also known as the point group $D_{4h}$, but the presence of an infinite number of axes of inversion makes $D_{\infty h}$ a more appropriate choice.</answer>
<answer>C) $D_{\infty h}$</answer>

---

# K   USE OF LARGE LANGUAGE MODELS

We used LLMs solely for proofreading and polishing text. All ideas, results, and arguments are the authors' own.

