# OpenReview forum: "Interleaved Reasoning for Large Language Models via Reinforcement Learning"
_ICLR.cc/2026/Conference — ICLR 2026 Conference Withdrawn Submission_

### Official Review · Reviewer_eReZ · 2025-10-30

**Soundness:** 2
**Presentation:** 3
**Contribution:** 2
**Rating:** 4
**Confidence:** 4

**Summary:**

This paper proposes Interleaved Reasoning (IR), an RL framework that enables LLMs to alternate between thinking and answering during reasoning. The approach introduces special <think> and <answer> tags and applies a conditional reward scheme to encourage correct intermediate answers while maintaining reasoning structure. Experiments across five reasoning datasets and three RL algorithms show consistent improvements in both reasoning efficiency and task accuracy, with notable reductions in time-to-first-token (TTFT).

**Strengths:**

- The paper addresses an important issue in reasoning LLMs — the trade-off between long, reflective reasoning chains and real-time efficiency — and provides a clear framework to investigate this trade-off.
- The method is easy to reproduce and compatible with standard RL algorithms, offering practical insights into the impact of intermediate rewards and structured reasoning patterns.
- The experiments covering multiple datasets, RL algorithms, and detailed ablations that help clarify the empirical effects of the proposed framework.

**Weaknesses:**

- Marginal technical novelty. The core mechanism of interleaved reasoning — introducing <think> and <answer> tags and training models to alternate between them — is relatively incremental. The method essentially reformulates the output format and applies a rule-based reward scheme within a conventional RL training setup.
- Potential conflict with self-reflection. Traditional think-then-answer reasoning allows the model to perform internal reflection and correction before producing a final response. The proposed interleaved approach may hinder such self-correction by encouraging the model to provide partial answers prematurely, promoting short-sighted reasoning and reducing the chance of global reassessment of reasoning paths.
- Limited evaluation of model adaptability. Although the paper compares different RL algorithms, it does not analyze how well the interleaved paradigm adapts across models with different reasoning characteristics. In particular, models like Qwen3, which already support explicit structured thinking, may behave differently, and this dimension is not sufficiently explored.
- Reward validation details. While a “format reward” is introduced to enforce the alternation between <think> and <answer>, the paper does not clearly describe how this formatting is verified or how to ensure the semantic distinction between thinking and answering segments. This raises concerns about the reliability of the reward signal.

**Questions:**

- The current experiments focus on relatively simple multi-hop reasoning tasks. Would the proposed method remain effective on more challenging benchmarks such as AIME 2024/2025 or similar high-difficulty reasoning datasets?
- Can the proposed interleaved reasoning framework be effectively applied to thinking models such as Qwen3, which already perform explicit reasoning-then-answering? Since Qwen2.5 reportedly “inherently possesses the ability to interleave,” could the observed gains simply reflect properties of the base model rather than the proposed RL scheme?
- In the reward design, the format reward depends on correctly distinguishing between <think> and <answer> segments. How is this verification implemented? How does the system ensure that the content inside <think> truly represents reasoning rather than a disguised partial answer?

---

> ### Author Response · Authors · 2025-12-04
>
> Dear Reviewer eReZ,
>
> Thank you for your constructive feedback and for recognizing the importance of our work in addressing the trade-off between reasoning efficiency and depth. We appreciate that you found our framework clear, reproducible, and our experiments extensive. We address your concerns and questions below:
>
> ---
>
> > **Marginal Technical Novelty**
>
> We appreciate this challenge, but we would like to emphasize the complexity of the underlying problem and the sophistication of our solution. While the interleaving structure appears simple, our core innovation lies in the **conditional reward strategy** and training dynamics that make it effective. The recent success of models like DeepSeek-R1, which utilize basic `<think>` and `<answer>` tags, further validates our approach; it demonstrates that straightforward structural modifications, when combined with proper training, yield substantial improvements.
>
> Beyond structural design, we offer several important findings regarding reward design and LLM reasoning dynamics: conditional intermediate rewards incentivize correct intermediate reasoning (Table 2), interleaved reasoning performance scales better with problem difficulty (Figure 3a), and correct answers are consistently shorter than incorrect ones (Figure 3c). These insights confirm that simplicity is a strength, enabling strong generalization and cross-domain adaptability.
>
> > **Potential Conflict with Self-reflection**
>
> You raise an important point regarding potential error accumulation from committal intermediate answers. Our results suggest that interleaved reasoning enhances efficiency by incentivizing models to reach correct intermediate conclusions *early*, thereby reducing the need for backtracking. While backtracking can improve reasoning, it often leads to overthinking and inefficiency. Our current results indicate that the structural benefits of interleaved reasoning outweigh potential committal disadvantages, achieving both higher accuracy and faster TTFT. However, we agree that investigating interactive feedback and correction within interleaved reasoning is a valuable direction for future work.
>
> > **Evaluation of Reasoning Models**
>
> Regarding the suggestion to test on reasoning models that "already support explicit structured thinking," we believe this would obscure the primary contribution of our work. Our goal is to evaluate the effectiveness of the **interleaved reasoning itself** when applied to base models. Testing on a model that already possesses long reasoning capabilities would make it difficult to isolate the gains attributable to our proposed method versus the model's inherent capabilities.
>
> > **Reward Validation and Verification**
>
> As described in **Section 3.2.2** and **Appendix D**, our training datasets (K&K and Musique) contain annotated intermediate ground truths. The reward $r_{intermediate}$ is strictly conditional: it is only awarded if the content within `<answer>` tags matches the ground truth sub-answer.
>
> The reward function $f(y_{answer}^{(k)})$ uses exact matching against these ground truth labels. If a model places reasoning "chatter" inside an `<answer>` tag, it will fail the match against the concise ground truth (e.g., "The Berlin Wall fell in 1989" vs. a long explanation), resulting in zero reward. This mechanism strictly enforces the semantic distinction between the thinking and answering tags during training.
>
> ---
>
> Sincerely,
>
> The Authors

---

### Official Review · Reviewer_1Qpr · 2025-11-01

**Soundness:** 2
**Presentation:** 2
**Contribution:** 2
**Rating:** 2
**Confidence:** 4

**Summary:**

The paper proposes training LLMs to alternate \<think\>\</think\> and \<answer\>\</answer\> with a conditional, rule-based reward for intermediate steps. The authors evaluate the proposed method on both in-domain (K&K, Musique) and out-of-domain (GPQA, MMLU, MATH) datasets.

**Strengths:**

- Clear motivation.

**Weaknesses:**

1. The definition and interpretation of the TTFT metric appear conceptually weak. In real applications, users primarily care about the final answer. For example, in Figure 1, the user’s actual need is to know the final “director,” not the model’s intermediate reasoning. If users are interested in the reasoning process, they can view the entire “think” chain, so interleaving does not inherently improve usability. Moreover, TTFT can be easily reward hacking: a model could output any meaningless statement early to artificially boost the metric. This undermines its validity as a measure of responsiveness.

2. The paper’s presentation of the intermediate reward mechanism is highly unclear. Although Equation (3) introduces the concept, it never specifies how correctness of intermediate steps is evaluated. The brief note that “f(y_answer^(k)) evaluates the answer correctness at step k” is insufficient and leaves readers guessing. Appendix D.2 defines a Correct() function, but again fails to explain the actual criterion for correctness. Determining intermediate correctness without ground truth is a well-known open challenge, and this work appears to sidestep it. The authors only declare that such a function exists, without revealing its logic, heuristics, or data sources. As a result, the core mechanism of this paper cannot be understood or reproduced.

3. The paper seems to sum all intermediate rewards and treat the result as a single final reward, according to Equations (3) and (5). If this is true, the model receives one identical reward signal across all tokens and steps, which defeats the goal of localized credit assignment. The approach would then be functionally equivalent to the traditional final-reward RL setup, with no genuine per-step feedback. The authors should clarify this point explicitly and explore using true step-level intermediate rewards, where each intermediate step receives its own immediate feedback.

4. The reported baseline results differ drastically from known benchmarks, casting doubt on the validity of the claimed improvements. In Tables 8 and 9 of the Qwen2.5 technical report, Qwen2.5-7B-Instruct reaches about 75.5 on MATH and 36.4 on GPQA, but this paper reports only 56.1 and 19.2. Similarly, Qwen2.5-1.5B-Instruct achieves 55.2 and 29.8 on those tasks respectively in the official report, whereas the paper reports only 30.8 and 6.6. Such large discrepancies cannot be attributed to random variation. The authors should reconcile these differences.

**Questions:**

See Weaknesses

---

> ### Author Response · Authors · 2025-12-04
>
> Dear Reviewer 1Qpr,
>
>
> Thank you for your detailed review and for recognizing the clear motivation behind our work. We address your concerns as follows:
>
> ---
>
> > **Validity and Utility of TTFT**
>
>
> We wish to clarify that TTFT (Time-To-First-Token, defined here as the first *answer* token) is a valid and useful metric for reasoning tasks for several reasons. First, as reasoning models evolve, inference times extend significantly, which can break the interaction flow in real-time AI applications. In these scenarios, users rarely have the bandwidth to examine lengthy reasoning traces. Interleaved reasoning addresses this by providing "incremental feedback," creating verifiable checkpoints that signal progress.
>
>
> Second, regarding reward hacking: our method explicitly prevents the model from outputting meaningless statements to boost TTFT via the **conditional reward mechanism**. The intermediate reward is *only* applied if the intermediate answer is **correct** relative to the ground truth. If a model outputs "meaningless statements," `Correct(y_answer)` evaluates to false, and no reward is given. Furthermore, the **Format Reward** penalizes incorrect structural tags. Thus, the model is incentivized to output *correct* partial answers early, not merely *any* tokens.
>
>
> > **Definition of Intermediate Correctness**
>
>
> To clarify the training process, we specifically utilize **Knights and Knaves (K&K)** and **Musique**. We chose these datasets precisely because they naturally contain subproblems with associated ground truths.
>
>
> The `Correct()` function referenced in Equation 3 and Algorithm 1 relies on a straightforward exact match (or normalized match) against these **existing intermediate ground truth labels** provided by the training datasets. A key contribution of our paper is demonstrating that while we *train* using these ground-truth-rich datasets, the model successfully *generalizes* to datasets without intermediate annotations (such as MATH and GPQA) during evaluation.
>
>
> > **Credit Assignment and Reward Summation**
>
>
> This is an excellent point regarding the nature of the rewards. We acknowledge the distinction between our approach and traditional PRM or ORM. As discussed in our Related Work (L125-127) and Appendix A, our method operates similarly to an ORM but grants partial credit for verified intermediate answers. We focus on a simple rule-based ORM to capture intermediate signals rather than incurring the complexity of training a separate PRM.
>
>
> > **Baseline Discrepancies**
>
>
> The discrepancies between our baselines and the Qwen2.5 technical report stem from differences in evaluation settings. The Qwen2.5 technical report uses 5-shot for GPQA and 4-shot for MATH, whereas our evaluation uses a **zero-shot** setting. Furthermore, instead of using the full dataset, we evaluated on the hardest subsets: the **Diamond** set for GPQA and the **Level-5** subset for MATH. These specifications are discussed in **Appendix F.2**. Consequently, the lower scores reflect the increased difficulty and lack of few-shot examples, not a deficiency in the baseline implementation.
>
> ---
>
> Sincerely,
>
> The Authors

---

### Official Review · Reviewer_xv6R · 2025-11-01

**Soundness:** 1
**Presentation:** 2
**Contribution:** 2
**Rating:** 4
**Confidence:** 4

**Summary:**

This paper proposes an RL-based paradigm intended to improve LLM reasoning efficiency by alternating between <think> (internal reasoning) and <answer> (public intermediate outputs). The authors introduce a conditional intermediate reward to encourage correct sub-answers, claiming reductions in TTFT and gain in Pass@1 across several reasoning datasets. Experiments are conducted on Qwen 2.5-1.5B/7B using PPO, GRPO, and REINFORCE++.

**Strengths:**

- The motivation of reducing TTFT while maintaining or improving reasoning quality is timely.
- The interleaved format is simple and compatible; the use of special tags (<think>, <answer>) is pragmatic.

**Weaknesses:**

- Training relies on K&K and Musique, both containing explicit sub-problem labels.

- Generalization to domains lacking intermediate supervision is claimed but not empirically tested.

- No ablation shows sensitivity to ε or to the relative weighting β in Eq. 2.

- No direct wall-clock latency or token-throughput numbers are reported.

- The experiments are conducted solely on the Qwen series models, without validation on other model families. In addition, the method has not been tested on models larger than 7B parameters.

- The proposed “conditional intermediate reward” essentially re-weights existing PRM signals based on batch accuracy; this appears to be an incremental engineering trick with limited contribution to the community.

**Questions:**

- Please provide variance across random seeds and confidence intervals for all Pass@1 results.

- In Figure 1, the “long” vs. “short” TTFT examples are described qualitatively. Could you quantify this difference with concrete numbers or timing statistics?

- Can you demonstrate actual wall-clock TTFT improvements under identical inference conditions?

- Does the model truly generalize without intermediate-ground-truth supervision?

- Ablate each term in Eq. (5): What happens if r_format or r_intermediate is removed?

---

> ### Author Response · Authors · 2025-12-04
>
> Dear Reviewer xv6R,
>
>
> We sincerely thank you for your review and for recognizing the timeliness of our motivation to reduce TTFT while maintaining reasoning quality. We also appreciate your feedback regarding the pragmatic design of our interleaved format. We address your concerns as follows:
>
> ---
>
> > **Generalization to Domains Lacking Intermediate Supervision**
>
>
> We would like to point out that we *have* tested the generalization to domains lacking intermediate supervision. As detailed in **Section 5.1** and **Table 2**, our models were trained *only* on K&K and Musique (which contain intermediate labels), but were evaluated on **MATH, GPQA, and MMLU**. These out-of-domain evaluation datasets **do not** contain intermediate ground truths. Additionally, as detailed in **Appendix E (Table 10)**, we conducted experiments on **GSM8K** *without any intermediate rewards* to further demonstrate the generalization capabilities of our method.
>
>
> > **Wall-clock Latency and Token Throughput**
>
>
> We utilize normalized TTFT (relative position) to ensure fair comparisons across different model outputs. As noted in **Section 5.2 (Reasoning Length Analysis)**, generation time scales linearly with token count in auto-regressive models. **Table 4** further demonstrates that our method reduces the *overall* response length by up to 37%. This reduction directly correlates to improved total throughput and reduced inference costs in wall-clock terms.
>
>
> > **Model Diversity**
>
>
> We selected the Qwen2.5 series (1.5B and 7B) as they represent the state-of-the-art open-weights models in their class. The fact that our method demonstrates consistent gains across two different model sizes (1.5B and 7B) and three different RL algorithms (PPO, GRPO, REINFORCE++) provides strong evidence of the method's robustness across architectures.
>
>
> > **Conditional Intermediate Reward**
>
>
> Thank you for the valuable feedback. Rather than viewing this as a "trick," we highlight that our conditional mechanism alleviates the credit assignment problem *without* the high computational cost and complexity of training a separate PRM. It provides a simple, rule-based alternative that achieves promising results. We believe this efficiency—achieving performance gains without complex auxiliary models—is a meaningful contribution to the community.
>
> ---
>
> Sincerely,
>
> The Authors

---

### Official Review · Reviewer_FNh6 · 2025-11-03

**Soundness:** 3
**Presentation:** 3
**Contribution:** 2
**Rating:** 4
**Confidence:** 3

**Summary:**

This paper introduces a novel training paradigm utilizing RL to encourage LLMs to adopt an interleaved reasoning strategy, which mixes thought steps and answer steps, for multi-hop question answering. The primary motivation is to mitigate the inefficiency and increased TTFT associated with traditional long reasoning. The authors demonstrate that this RL-based approach can significantly reduce latency while maintaining or improving performance under certain reward schemes.

**Strengths:**

1. The most interesting result is the significant reduction in the TTFT. This directly addresses a critical and practical limitation of using CoT in real-world applications and demonstrates the potential for faster, more efficient reasoning in LLMs.

2. The paper is generally well-structured and easy to follow. The methodology and experimental setup are clearly explained, which aids in the comprehension of the proposed interleaved reasoning framework.

**Weaknesses:**

1. A major concern is that the interleaved reasoning appears weak when not supported by dense intermediate rewards. The performance of the model in the setting without strong intermediate reward signals is often suboptimal or even worse than standard baselines, suggesting that the RL framework alone is not sufficient to reliably improve reasoning quality.

2. While the model's performance significantly improves when strong intermediate rewards are introduced, this outcome is largely expected as the model receives a much stronger signal than standard end-to-end RL or supervised methods. The current comparison to baselines in this heavily-rewarded setting is therefore not entirely fair or informative. A more rigorous comparison is needed against models that also leverage similar process-level supervision or intermediate reward signals.

**Questions:**

1. Why does the proposed interleaved method, without the aid of dense intermediate rewards, struggle to maintain or improve performance compared to baselines? What architectural or training adjustments could be made to improve the robustness of the core interleaving mechanism?

2. Could the authors perform a comparison against representative baselines that also incorporates process rewards or intermediate supervision? This would significantly strengthen the claim that the interleaved structure is the source of the gain, rather than just the dense reward signal.

---

> ### Author Response · Authors · 2025-12-04
>
> Dear Reviewer FNh6,
>
>
> We thank you for your thoughtful review and for recognizing our work as a "novel training paradigm" that addresses the critical issue of latency in reasoning LLMs. We appreciate your positive remarks regarding the "significant reduction in TTFT" and the clarity of our methodology. We address your concerns as follows:
>
> ---
>
> > **Performance without Dense Rewards**
>
>
> This is a great observation. However, we would like to highlight that the performance of the base `Interleave` method (without intermediate rewards) is comparable to the `Think-Answer` baseline across several datasets, while delivering a *massive gain in efficiency*.
>
>
> A primary contribution of this work is that the base `Interleave` model achieves this comparable accuracy while **reducing TTFT**, which represents a significant efficiency gain. We will expand on this discussion in the main text to clarify this trade-off.
>
>
> > **Process Supervision**
>
>
> We agree that intermediate rewards provide a stronger signal; however, our goal is to demonstrate that Interleaved Reasoning is a simple yet effective alternative to standard Outcome Reward methods, offering the unique benefit of significantly lower TTFT. As discussed in **Appendix A**, we emphasize that avoiding reliance on a complex Process Reward Model (PRM) is an advantage of our approach, prioritizing simplicity and efficiency.
>
>
> Furthermore, to address the concern regarding the necessity of dense supervision, we explicitly evaluated the robustness of the interleaved structure without these rewards. As detailed in **Appendix E (Table 10)**, we conducted experiments on **GSM8K** *without any intermediate rewards*. This demonstrates that the model performs well and maintains robustness based on the interleaved structure itself, proving that the efficiency gains (lower TTFT) are achieved without relying solely on dense process supervision.
>
> ---
>
> Sincerely,
>
> The Authors

---

### Note · Authors · 2025-12-04

I have read and agree with the venue's withdrawal policy on behalf of myself and my co-authors.